# Investigation of Forest Fire Characteristics in North Korea Using Remote Sensing Data and GIS

**Ri Jin** [1,2] and **Kyoo-Seock Lee** [3,*]

1 College of Geography and Ocean Sciences, Yanbian University, Hunchun 133300, China
2 Northeast Asian Research Center of Transboundary Disaster Risk and Ecological Security, Yanbian University, Hunchun 133300, China
3 Department of Landscape Architecture, Graduate School, Sungkyunkwan University, Suwon 16419, Korea
* Correspondence: leeks@skku.edu; Tel.: +82-31-290-7845

**Abstract:** Forest fires cause damage to property and the environment around the world every year. North Korea has suffered from fires every year. Fires may lead to temporary or permanent damage to forest ecosystems, long-term site degradation, and alteration of hydrological regimes, producing detrimental impacts on economies, human health, and safety. In North Korea, fires cause serious damage to the affected mountainous environment. However, it is very difficult to obtain ground information or perform field checks because of the political isolation of North Korea. Thus, there are few studies that have investigated North Korean fires. In this situation, remote sensing techniques and digital topographic data can be used to investigate fire characteristics in North Korea. In this study, fire trends were analyzed using Moderate Resolution Imaging Spectroradiometer (MODIS) data from the Land Processes Distributed Active Archive Center (LPDAAC) from 2004 to 2015, and Landsat data were processed to estimate burned areas in South Hamgyong Province (SHP) and Gangwon Province (GWP) in North Korea. The burn severity of large fires in elevation, slope, and landform features was also analyzed to investigate large fire-burned areas using 30-m-resolution Global Digital Elevation Model (DEM) data from the United States National Aeronautics and Space Administration (NASA). After the results were compared and discussed, the following conclusions were derived. (1) In terms of location, fires in SHP were relatively concentrated along BaekDu-DaeGan (BDDG), while fires in GWP were scattered throughout the province. (2) In terms of size, the large fire-burned areas with an area greater than 1000 ha are significantly more frequent in SHP than in GWP. In brief, large fires occurred more frequently and were more serious in SHP than in GWP. (3) In terms of forest type, coniferous areas were more susceptible to damage from fires and large fires than deciduous areas in both GWP and SHP. This is attributed to the combustible resin within the coniferous trees. Particularly, when a crown fire occurs, it tends to spread rapidly throughout the coniferous forest. (4) Regarding landforms, most large fires occurred along windward-side open slopes, while there were very few fires in shallow valleys, high ridges, or U-shaped valleys. It is believed that cultivation in high-elevation terrain and a lack of fire-extinguishing equipment and systems allow large fires to spread quickly. North Korea is very susceptible to large fire damage and must develop preparation measures against such situations.

**Keywords:** forest fire; burn severity; remote sensing; inaccessible region

## 1. Introduction

As an important ecological factor in the forest ecosystem, forest fires have a positive impact on the regeneration and succession of tree species. However, excessive fires have a major impact on societies, economies, and ecological environments and seriously threaten the lives and social stability of the many inhabiting organisms [1,2]. Fires profoundly interact with ecosystems, directly affect energy flow, and force the transformation of the

whole forest ecosystem. During combustion, large amounts of $CO_2$, $CH_4$, $NH_3$, $NO_x$, and CO gases are released into the atmosphere [3,4]. The structure, function, and processes of forest ecosystems are also altered, which has important impacts on regional and global ecological security [5,6].

In general, most fires in North Korea (North Korea) are attributed to an increase in the residential slash-and-burn method of field management [7]. In North Korea, fires frequently occur across the entire region, with forests comprising about 80% of the country [8], except in the western plains. In May of 2004, massive fires occurred throughout North Korea. According to the Earth Fire Monitoring Center in Germany, by mid-May alone, 170 fires occurred in North Korea [9]. The fires predominately occurred in remote mountainous regions in the spring and spread rapidly due to dry conditions, strong winds, and little rain. Several massive fires developed, resulting in heavy damage to forest resources due to a lack of equipment and personnel mobilization for management. However, data on North Korean fire damage have not been made public because of the country's political and economic isolation, which prohibits the collection of ground information and the performance of field checks.

In Korea, various studies have investigated fire damage using satellite imagery, such as that of Landsat Thematic Mapper (Landsat TM) and Landsat Enhanced Thematic Mapper Plus (Landsat ETM+) [10–17], IKONOS [18], LiDAR [19], SPOT-4 [20], and MODIS [21]. However, all studies have focused on fires in South Korea.

There has been no research focused on North Korean fires because of data unavailability and the inaccessibility of the region. However, the North Korean government does not have the capacity to respond effectively to the recent increase in fire damage [22]. Thus, it is necessary to investigate burned areas, especially to determine the spatio-temporal pattern of fires in North Korea. Therefore, the purpose of this study is to evaluate the spatial characteristics of North Korean burned areas quantitatively using multi-source remote sensing data. These findings can be used for restoring damaged forests in North Korea after reunification.

## 2. Materials and Methods

### 2.1. Study Site

North Korea occupies the northern portion of the Korean Peninsula, bordered by the northern part of the Demilitarized Zone (DMZ) [23] and comprising an area of 123,138 km$^2$ covering 55% of the Korean Peninsula (222,784 km$^2$). The geographic location of North Korea is from latitude 37°41′00″N to 43°00′36″N and longitude from 124°18′41″E to 130°41′32″E. North Korea shares land borders with China across the Yalu River, Russia, across the Tumen River to the north, and borders South Korea along the DMZ. To its west are the Yellow Sea and Korea Bay, and to its east lies Japan across the East Sea of Korea (Figure 1).

About 80% of North Korea's terrain is mountains and highlands, which are separated by deep and narrow valleys. All mountains of the Korean Peninsula with an elevation of 2000 m or more are located in North Korea. Forest covers over 80% of the country, mostly on steep slopes. North Korea has two distinct mountain ranges; the Korea Divide called BaekDu-DaeGahn (BDDG), which extends from Mt. Baekdu (Changbai) to Mt. Jiri in South Korea, and the Nahngrim Mountains from north to south, which separate Hamgyong Province from Pyongan Province. GWP and SHP are located to the south of BDDG and have been damaged by large fires in North Korea. Therefore, GWP and SHP were selected for the investigation of spatial characteristics of burned areas in this study (Figure 2).

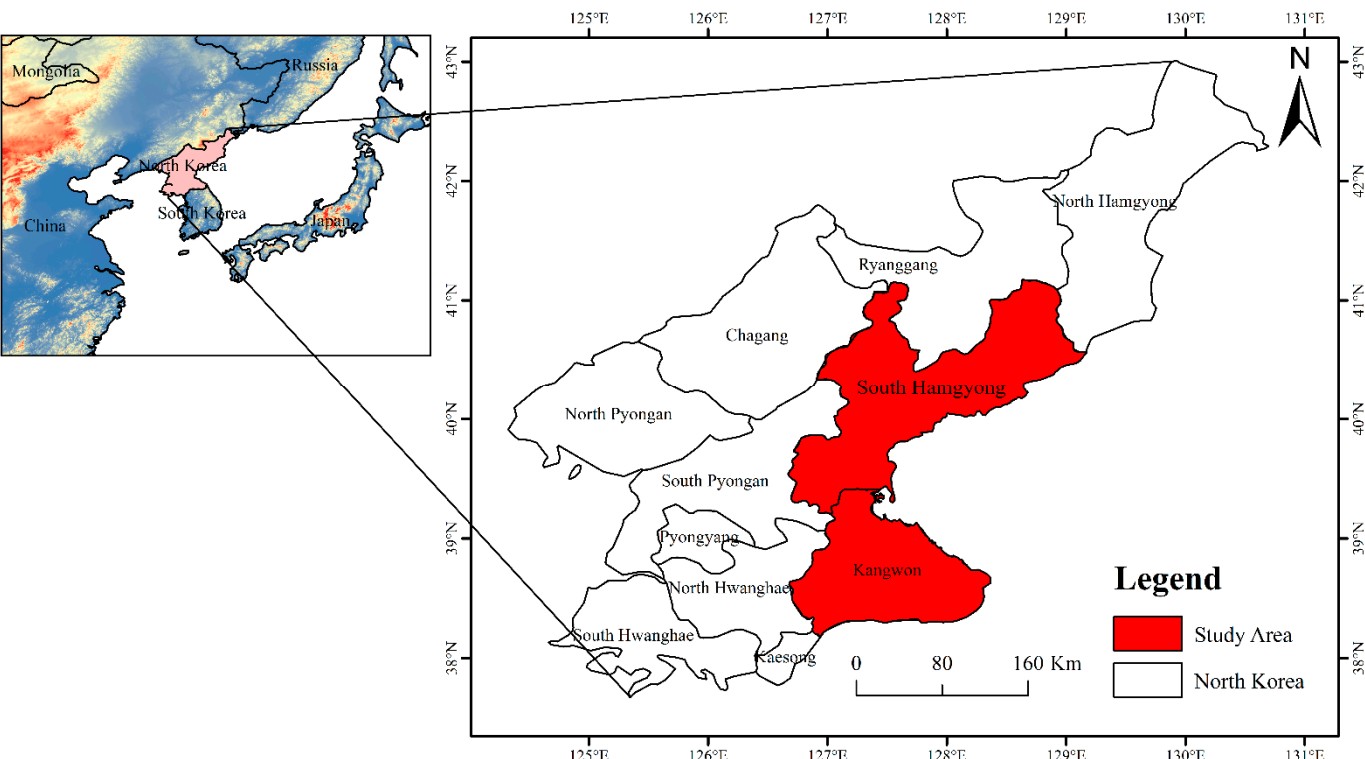

**Figure 1.** Study area.

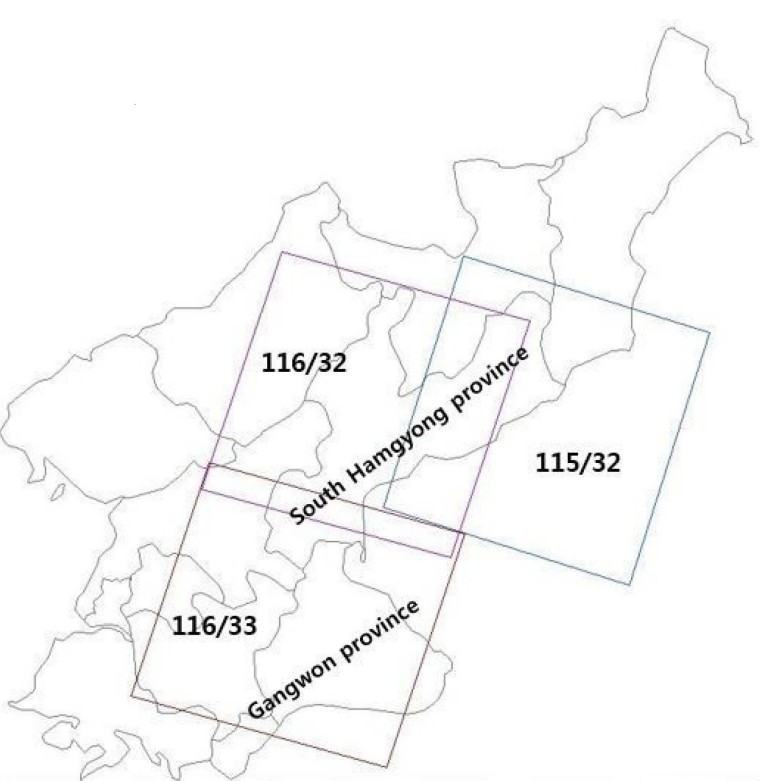

**Figure 2.** The path and row numbers of Landsat orbits that cover GWP and SHP.

North Korea has a continental climate with four distinct seasons [24]. Most of North Korea is classified as a continental climate with humid, hot summers and cold, dry winters. The daily average high and low temperatures are −3 and −13 °C in January and 29 and 20 °C in August. In the summer, there is a short rainy season called 'Changma'. Long

winters bring bitter cold and clear weather interspersed with snowstorms as a result of northern and northwestern winds that blow from Siberia. Summer tends to be short, hot, humid, and rainy because of the southern and southeastern monsoon winds that bring moist air from the Pacific Ocean. Spring and autumn are transitional seasons marked by mild temperatures and variable winds. On average, approximately 60% of all precipitation occurs from June to September. The annual average precipitation is about 1500 mm, and the precipitation gradually decreases from south to north. Natural hazards include late spring droughts that are often followed by severe flooding and typhoons, causing serious damage to North Korea at least once every summer or early autumn on average [25].

### 2.2. Data Used

The data (Table 1) used in our study to analyze the fire characteristics in North Korea were remote sensing data (Landsat image, MODIS fire production) and GIS data (DEM, fire origination point, land use, and digital administrative map of North Korea).

**Table 1.** Data used in this study.

| Data Type | | Date | Resolution | Path/Row | Source |
|---|---|---|---|---|---|
| Landsat | GWP | 2008/04/20 (forest type)<br>2003/06/01 (pre-fire)<br>2004/06/03 (post-fire)<br>2007/04/09 (pre-fire)<br>2009/06/01 (post-fire)<br>2014/05/30 (pre-fire)<br>2015/05/17 (post-fire) | 30 m | 116-33<br>116-33<br>116-33<br>116-33<br>116-33<br>116-33<br>116-33 | U.S. Geological Survey (USGS) (http://earthexplorer.usgs.gov/, accessed on 16 September 2021) |
| | SHP | 2008/04/20 (forest type)<br>2008/04/03 (forest type)<br>2002/05/05 (pre-fire)<br>2005/05/29 (post-fire)<br>2010/06/04 (pre-fire)<br>2011/05/14 (post-fire)<br>2014/05/30 (pre-fire)<br>2015/05/01 (post-fire)<br>2015/05/10 (post-fire) | | 115-32<br>116-32<br>116-32<br>116-32<br>116-32<br>116-32<br>116-32<br>116-32<br>115-32 | |
| MODIS | MOD14A1 | 2004–2015 | 1 km | 27/4, 27/5 and 28/5 | LPDAAC (http://modis.gsfc.nasa.gov/, accessed on 16 September 2021) |
| Geographic Information System (GIS) | Aster Global DEM | | 30 m | | USGS (http://earthexplorer.usgs.gov/, accessed on 16 September 2021) |
| | Fire occurrence point | 2004–2015 | | | Fire Information for Resource Management System (FIRMS) (https://www.earthdata.nasa.gov/, accessedon 16 September 2021) |
| | Land use | 2010 | 30 m | | GlobeLand30: Global Geo-information Public Product (http://www.globallandcover.com/, accessed on 16 September 2021) |
| | Digital administrative map of North Korea | | | | National Geographic Information Institute (https://www.ngii.go.kr/, accessed on 16 September 2021) |

### 2.3. Study Procedure

Figure 3 shows the study procedure for investigating the characteristics of fires in North Korea in this study.

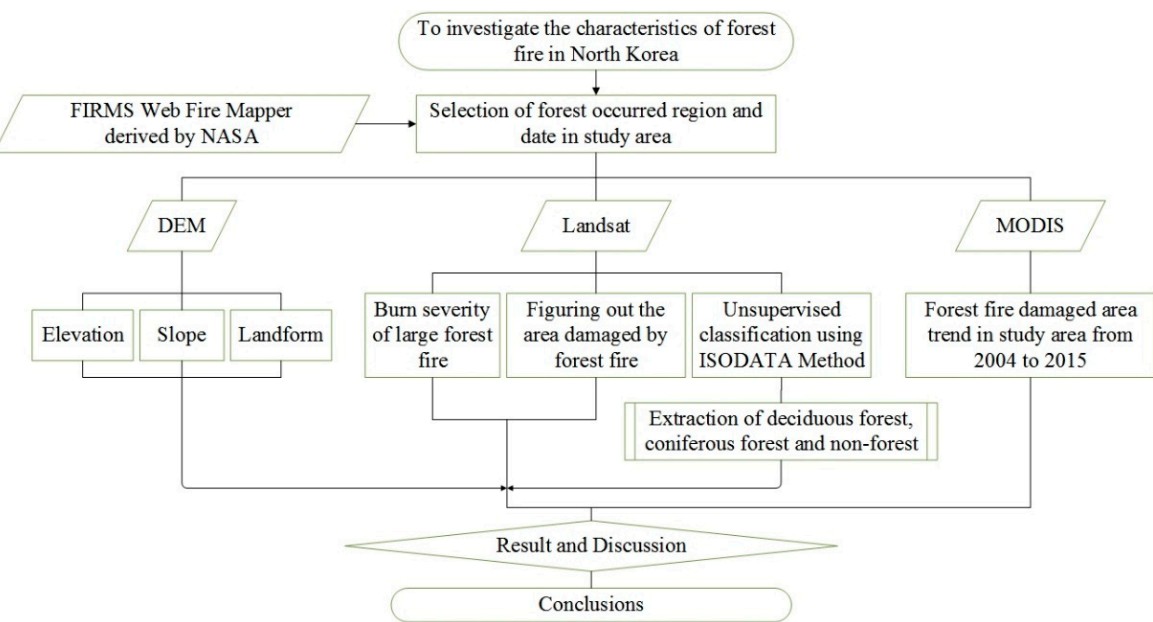

**Figure 3.** Flow chart of the procedures used in this study.

Firstly, the date and frequency of fires in North Korea were derived from the FIRMS Web Fire Mapper produced by the United States National Aeronautics and Space Administration (NASA). Available satellite images were retrieved and downloaded from the United States Geological Survey (USGS).

Secondly, fire trends from 2004 to 2015 were analyzed using MODIS data from the LPDAAC.

Thirdly, Landsat data were processed for estimating the burned area.

Fourthly, to investigate burned areas, the burn severity of large fires in elevation, slope, and landform features was analyzed using 30-m-resolution Global Digital Elevation Model (DEM) data from NASA for the selected year.

Fifthly, for discriminating forest type, unsupervised classification was performed using the Iterative Self-Organizing Data Analysis Technique (ISODATA) because training data selection is not easy due to the shortage of ground data in the study site.

Finally, results are discussed, and conclusions are derived.

*2.4. Methods*

2.4.1. Selection of Fire Occurrence Years and Fire Trend Analysis

To select the fire occurrence year, the annual variation in the number of fire hot spots in the forest area was collected from NASA FIRMS data in combination with the 30 m resolution land use data provided by GlobeLand30: Global Geo-information Public Product. Figure 4 shows the number of hot spots that occurred in the study site from 2004 to 2015. The location and magnitude of each burn scar were determined using remote sensing data because North Korea is currently inaccessible. In addition, 2004, 2005, 2009, 2011, 2014, and 2015 show more hot spots than in other years, indicating a higher frequency and average magnitude of fires than in other years. The frequency shown in Figure 4 represents total fire occurrence because FIRMS records all hot spots, even if the burn scar is very small. Therefore, burn scars were assessed using band composition (Red: Short Wave Infrared (SWIR), Green: Near Infrared (NIR), Blue: Visible Red) for six years of Landsat images. As a result, the fires of 2004, 2009, and 2015 in GWP appeared clearly in Landsat images, as did those of 2005, 2011, and 2015 in SHP. Therefore, Landsat images from 2004, 2009, and 2015 in GWP and of 2005, 2011, and 2015 in SHP were selected for investigation. As fires occurred most frequently in the spring, fires in that season in GWP and SHP were selected for investigation.

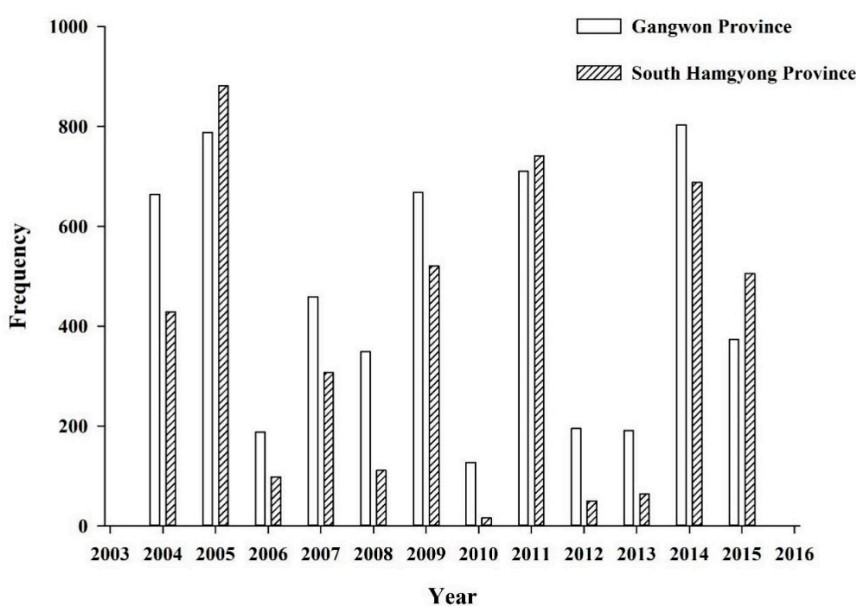

**Figure 4.** Number of hot spots derived from MODIS data from 2004 to 2015 in both study areas (Unit: frequency; Source: NASA FIRMS).

For analyzing fire trends, the Terra MODIS Thermal Anomalies/Fire product MOD14A1 (Table 1) was used. Data were derived primarily from MODIS 4- and 11-micrometer radiance values, with path/row numbers of Landsat orbits of 27/4, 27/5, and 28/5 from LPDAAC. The images were acquired online from LPDAAC (Table 1). Additionally, geometric corrections were performed with Universal Transverse Mercator (UTM) map projection and World Geodetic System (WGS) data. Furthermore, the three images were mosaicked to create one image for extracting burned areas in the study site.

The fire detection strategy is based on the absolute detection of a fire with sufficient strength and with good detection relative to the background (to account for the variability of surface temperature and reflection by sunlight). The outputs include fire occurrence (day/night), fire location, logical criteria used for fire selection, and detection confidence and distinguish fire, no fire, and no observation conditions. MOD14A1 data are produced every 8 days at a 1-km resolution as a gridded level 3 product in a sinusoidal projection. These data are unique in that they have three dimensions: fire-mask (1D) and maximum fire-radiative-power (2D) are provided for each day (3D) in the 8-day period, and the MOD14A1 fire-mask data includes 10 classes (Table 2).

**Table 2.** MOD14A1 fire-mask classes (Source: LP DAAC).

| Value | Class |
|---|---|
| 0 | missing input data |
| 1 | not processed (obsolete) |
| 2 | not processed (obsolete) |
| 3 | water |
| 4 | cloud |
| 5 | non-fire |
| 6 | unknown |
| 7 | fire (low confidence) |
| 8 | fire (nominal confidence) |
| 9 | fire (high confidence) |

Pixel values 8 and 9, with respective nominal and high confidence (Table 2), were chosen from the MOD14A1 data, and burn scars were extracted to calculate the area and investigate the fire trend from 2004 to 2015 in the study area.

### 2.4.2. Estimation of Burned Area

To estimate burned area in GWP, Landsat data with respective path/row values of 116/33 were used because they cover most of GWP. However, SHP was not covered by a single Landsat image but required three images with respective path/row values of 116/32 (main image), 116/33, and 115/32 (Figure 2). These images were processed to delineate burned areas (Table 1). The Landsat ETM+ 116/32 image on 30 May 2014 was used because it is larger than the other images and covers the whole burned area in one image.

We selected Level 1T images in this study because of their geometric accuracy and used the DEM for topographic accuracy. The geodetic accuracy of the product depends on the accuracy of the ground control points, and the resolution of the DEM used (http://landsat.usgs.gov/descriptions_for_the_levels_of_processing.php, accessed on 16 September 2021).

For detecting burned areas, the Normalized Burn Ratio (NBR) was calculated for both GWP and SHP. Analysis of change in the spectral pattern due to fire can be used for detecting burn scars, as the index was formed through a combination of various spectral bands, including NBR [26], and was calculated as:

$$NBR = \rho NIR - \rho MIR / \rho NIR + \rho MIR \tag{1}$$

where $\rho NIR$ is the reflectance in the Near Infrared (NIR), and $\rho MIR$ is the reflectance in the mid-infrared.

The NBR index is able to differentiate burned areas from other land uses except bare fields and crops [27]. The options considered to eliminate unwanted noise were the application of thresholds or masks. Following this, NBR data were used to auto-digitize the burn scars and modify the unwanted areas to produce the final burn scar map.

### 2.4.3. Discriminating Forest Type in Burned Areas

For discriminating forest type, unsupervised classification using the widely used ISODATA clustering algorithm was used because training data selection is limited due to the shortage of ground data [28]. The forest type (coniferous or deciduous) was classified using the 116/33 image on 3 April 2008, 116/32 image on 12 October, and 115/32 image on 20 April 2008 (Table 1). Images from 2008 were used to classify forest types because of the low fire frequency that year. The images were acquired online from the USGS (Table 1). ISODATA is self-organizing and requires relatively little human input. A sophisticated ISODATA algorithm normally requires the analyst to specify seven criteria of Cmax, T, M, minimum members in a cluster (%), maximum standard deviation (max), split separation value, and minimum distance between cluster means, C [29].

In this study, the minimum number of classes in the ISODATA algorithm was set to 10, the maximum number of classes to 30, the maximum iterations to 30, the maximum standard deviation to 5, and the minimum distance to 3. Furthermore, the area of forest type damaged by fire was investigated using GIS data.

### 2.4.4. Selection of Large Fire

After detecting the burn scar by FF in 2004, 2009, and 2015 (GWP) and 2005, 2011, and 2015 (SHP), the patch area was calculated for every year, and fire scars with areas of more than 5000 ha in GWP and more than 10,000 ha in SHP were chosen. As a result, there is one region whose area was more than 5000 ha on 16 April 2004 (GWP) and three regions whose area was more than 10,000 ha on 3 May 2005, 13 April 2011, and 24 April 2015, respectively (SHP). Therefore, these dates of large fires were selected for investigation.

### 2.4.5. Estimation of Burn Severity

NBR and Differenced Normalized Burn Ratio (dNBR) can be used to effectively estimate burn severity from a fire from remote sensing data [30–33] and show the best ratio between burn severity ranges [12]. The NBR was originally developed for use with Landsat TM and ETM+ bands 4 and 7, but it will work with any multispectral sensor

(including Landsat 8) with an NIR band between 0.76–0.9 μm and a SWIR band between 2.08–2.35 μm [34]. The NBR is temporally different between pre-fire and post-fire datasets to determine the extent and degree of change caused by burning.

Since fire effects on vegetation produce a reflectance increase in the mid-infrared (MIR) spectral region and an NIR reflectance decrease [35,36], bi-temporal image differencing is frequently applied to pre-fire and post-fire NBR images, resulting in dNBR [31]. The dNBR was calculated by subtracting post-fire NBR from pre-fire NBR:

$$dNBR = NBRprefire - NBRpostfire \qquad (2)$$

The dNBR data range is from −2 to 2. To facilitate burn severity analysis, dNBR data were linearly transformed to 8-bit integer data ranging from 0 to 255. In addition, the dNBR data were classified into 6 classes (Unburned, Low, Moderate, High, Very High, and Extreme) according to the formulae in Table 3.

**Table 3.** Burn severity classes (μ: Mean; δ: Standard deviation, Source: KFRI, 2013).

| Class | Interval |
|---|---|
| Unburned | $x < \mu - 2\delta$ |
| Low | $\mu - 2\delta <= x < \mu - 1\delta$ |
| Moderate | $\mu - 1\delta <= x < \mu$ |
| High | $\mu$ |
| Very High | $\mu <= x < \mu + 1\delta$ |
| Extreme | $\mu + 1\delta <= x < \mu + 2\delta$ |

2.4.6. Investigation of Topographic Characteristics

To investigate the topographic characteristics in burned areas, elevation, slope, and landform were analyzed. The landform thematic map was produced using DEM data and was based on the Topographic Position Index (TPI) [37]. The DEM data from the USGS were used for analyzing the elevation and landform in burned areas. In addition, the number of hot spots of fire data (2004–2015) from NASA was used to estimate the frequency and magnitude of fires by year and season in GWP and SHP (Table 1). A digital administrative map of North Korea from the National Geography Institute in Korea was used for masking the Landsat images in both provinces (Table 1).

This study analyzed circular neighborhoods of type small, with a 30 m unit, and large, with a 120 m unit, because the DEM cell size is 30 m. Furthermore, 10 landform classes were created (1. Canyons; 2. Hills in valleys; 3. Shallow valleys; 4. Mid-slope ridges; 5. High ridges; 6. Open slopes; 7. Plains; 8. U-shaped valleys; 9. Highland drainages; 10. Upper slopes) using small- and large-neighborhood Topographic Indexes (TPI). The landform characteristics in burned areas were investigated using GIS.

## 3. Results
### 3.1. Temporal and Spatial Distribution Characteristics of Fires
3.1.1. Fire Temporal Distribution Characteristics

Figure 5 shows the fire trends in the study area from 2004 to 2015. The areas with values of 8 or 9 in GWP and SHP were selected to investigate fire occurrence from 2004 to 2015 (Figure 5). As in Figure 5, after summing the burned areas of GWP and SHP, the highest occurred in 2005, with 74,420 ha, followed by 2011 (68,930 ha), 2004 (56,160 ha), 2014 (48,200 ha), 2009 (46,360 ha), and 2015 (36,430 ha). The total amount of GWP affected by fire was 219,500 ha, and that of SHP was 215,650 ha; burned area trends in the two study areas are significantly different from 2004 to 2015. The trends of fire area in GWP and SHP are clearly opposing (GWP shows a downward trend, while SHP shows an upward trend); however, the overall trend is upward. This indicates that the increasing burned area of SHP contributes more to overall fires than those of GWP. Therefore, large-scale fires in SHP are the main reason for increasing fires in the study area.

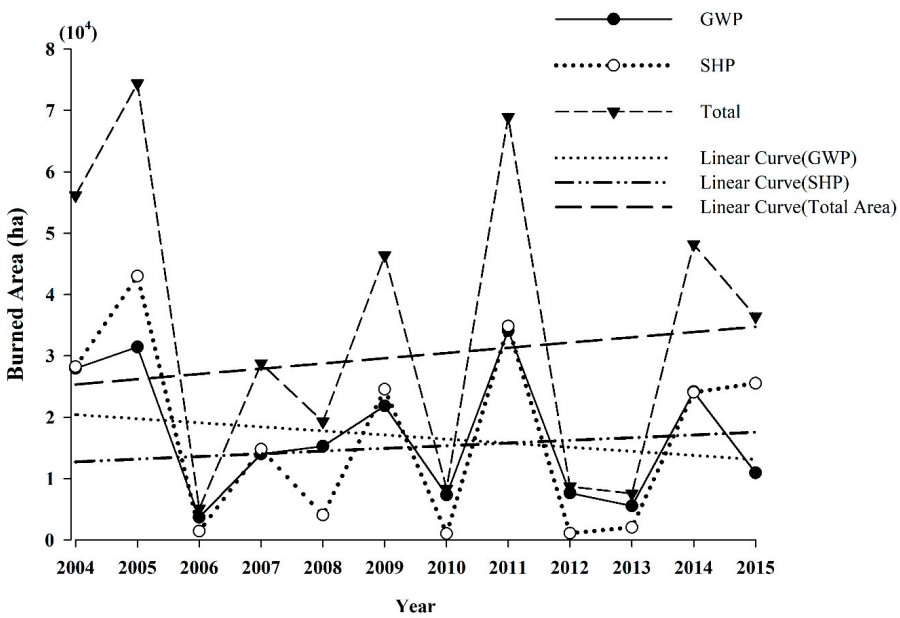

**Figure 5.** Burned areas (MODIS) in the study regions from 2004 to 2015.

### 3.1.2. Fire Spatial Distribution Characteristics

The NBR was calculated to derive the burned area in the study area. Figure 6a,c shows the spatial distribution of burned area in GWP and SHP. In terms of location, fires in 2004, 2009, and 2015 in SHP occurred along BDDG and were relatively concentrated, while those in GWP were scattered throughout the province (Figure 6a,c). In terms of scale, fires in SHP were larger than those in GWP. The differences between the two provinces in fire location and size were significant.

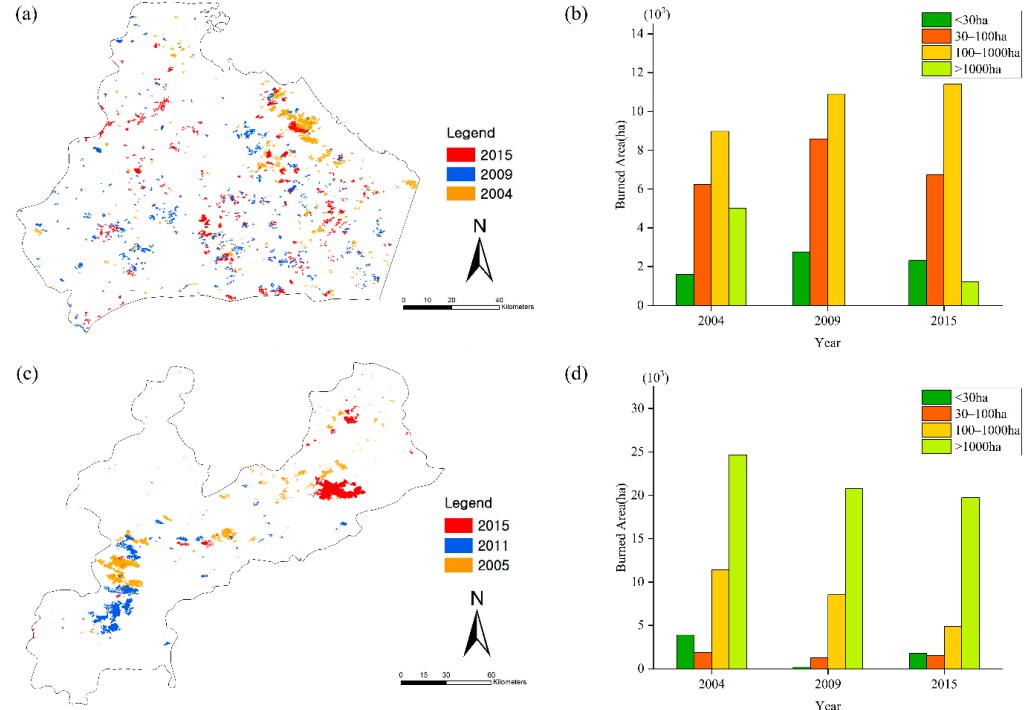

**Figure 6.** Burned areas in each study site. (**a**) Spatial distribution of burned areas in GWP; (**b**) Burned areas in GWP for three years (Unit: ha); (**c**) Spatial distribution of burned areas in SHP; (**d**) Burned areas in SHP in three years (Unit: ha).

The derived burned areas for each year were classified into four categories in terms of area: less than 30 ha, 30 ha to 100 ha, 100 ha to 1000 ha, and more than 1000 ha (Figure 6b,d).

Of the three studied years, fires in 2009 produced the largest burned area of 22,242.4 ha in GWP, followed by those in 2004 (21,846.5 ha) and 2015 (21,675.5 ha). The total burned area was 65,764.4 ha for the three years. The fires in 2005 showed the largest burned area of 41,811.9 ha in SHP, followed by those in 2011 (30,758.8 ha) and 2015 (27,888.7 ha). The total burned area was 100,459.4 ha for the three years and was 34,695 ha greater than that of GWP. The burned area in SHP contains more agricultural area than in GWP, as many agricultural fields are located in mountainous areas.

In terms of burned area, GWP had more areas less than 30 ha (<30 ha), 30–100 ha, and 100–1000 ha compared to SHP. However, the number of burned areas greater than 1000 ha in SHP was significantly larger than in GWP. Thus, even if fires occurred more frequently in GWP than in SHP for the three years, there were more fires per km$^2$ in SHP, and those fires tended to be larger in size.

*3.2. Large Fires Burn Severity Extraction Results*

To investigate large fire-burned areas, areas greater than 5000 ha in the 2004 fires (from 16 to 18 April) in GWP and those greater than 10,000 ha in the 2005 fires (from 3 to 5 May), 2011 fires (from 9 to 13 April), and 2015 fires (from 24 to 27 April) in SHP were selected. Pre-fire/post-fire Landsat images were chosen to extract the fire severity (Figure 7).

Though burn severity has different spatial distribution characteristics in the four burned areas, the Uiwon large fire was the largest. A very high burn severity level dominated in this large fire, and most of the burned area was impacted severely (Figure 7D). Each spatial region was segmented into many patches. The burned areas were very widely distributed, and the largest fire crossed the river. The results show that a moderate burn severity level was dominant in the Gaehak large fire in GWP (Figure 7A) and in the Jeong-pyong large fire in SHP (Figure 7C). Moderate and high burn severity levels dominated in the Hamju large fire (Figure 7B).

Burn severity was classified into six categories (Table 3) and then into three classes "Un-burned and low" (unburned, low, and moderate), "Moderate" (high), and "Extreme" (very high and extreme) [38]. Figure 7 shows that the "Unburned and low" burn severity level was dominant, and "Extreme" burn severity occurrence, such as in crown fires, was least frequent. However, the Uiwon large fire was significantly different from the other three large fires (Figure 8), with the largest burn severity level of "Extreme" (54.2%), followed by "Moderate" (37.5%) and "Unburned and low" on the windward-side. More than half the large fire-burned area was destroyed by fire. Uiwon was not only the largest burned area but also showed the most serious burn severity of the four large fires in this study.

For comparison, the Samcheok large fire on 7 April 2000, in Samcheok-Si, South Korea, was compared with large fires in North Korea because the burned area was the same as large fires in this study at 16,201 ha. Both fires occurred in eastern coastal regions of the Korean Peninsula, and the two are the most similar large fires in the two countries during the study period. In the Samcheok large fire, the largest burn severity level was "Unburned and low" (34.9%), followed by "Moderate" (33.5%) and "Extreme" (31.6%) [37]. The results show that the Samcheok large fire damage was more serious than that in Gaehak, Hamju, and Jeongpyong but less serious than that of the Uiwon large fire.

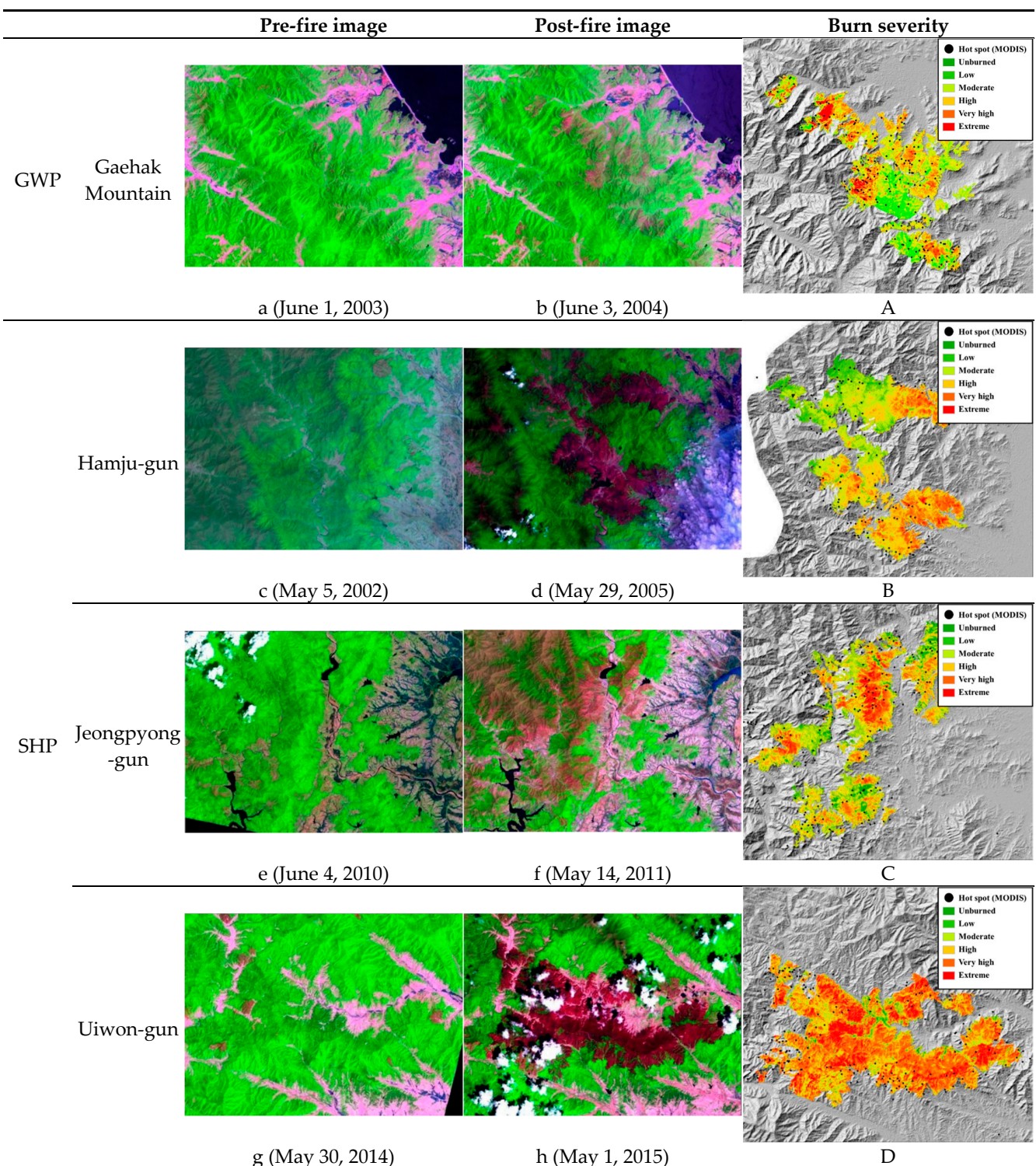

**Figure 7.** Burn severity maps of large fires. (**A**) The Gaehak large fire; (**B**) The Hamju large fire; (**C**) The Jeongpyong large fire; (**D**) The Uiwon large fire.

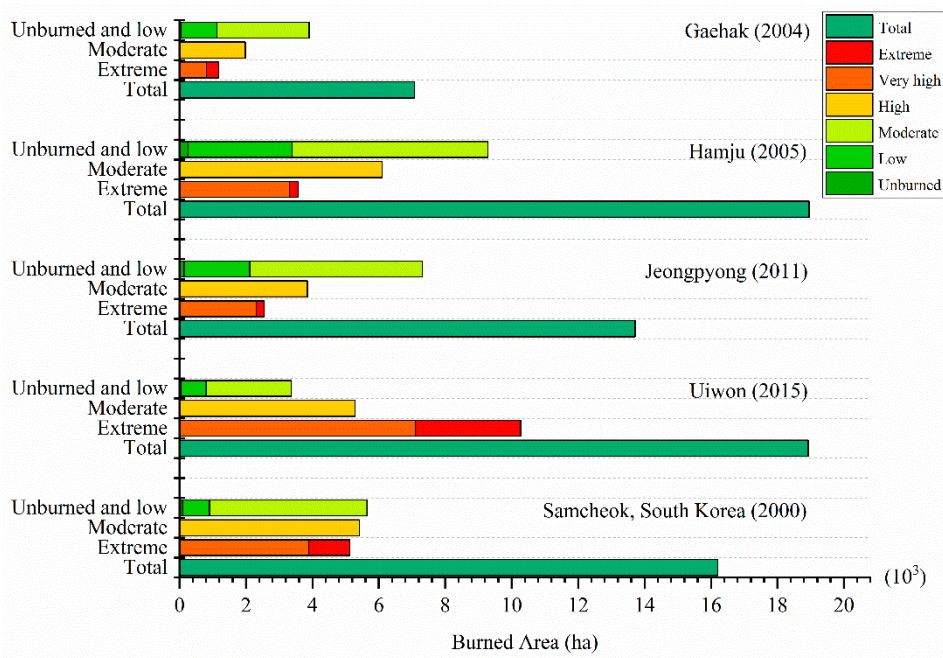

**Figure 8.** Burn severity assessment by dNBR of burned areas in the two study areas.

### 3.3. Forest Type and Topographic Features of Burned Areas

3.3.1. Forest Type

Forest vegetation is an important part of the global forest ecosystem. An accurate understanding of the types and spatial distributions of forest tree species is an important prerequisite for the accuracy of forest monitoring indicators, such as forest area and spatial location dynamics [39]. To estimate the forest type damaged by fires, a forest map was derived using Landsat images from 20 April 2008 in GWP and from 20 April and 3 April 2008 in SHP.

As shown in Figure 9a, the forests in GWP are mostly composed of coniferous forests and deciduous forests in BDDG, and coniferous forests were more prevalent than deciduous forests. The non-forested areas, which include agricultural areas, residential areas, and bare soil, were distributed in coastal and low-elevation areas. While most coniferous forests were distributed along BDDG in SHP (Figure 9b), with an elevation greater than 1400 m, deciduous forests were mainly distributed in mountainous areas with elevations ranging from 400 m to 1400 m. Furthermore, non-forested areas were distributed in coastal and low-elevation areas.

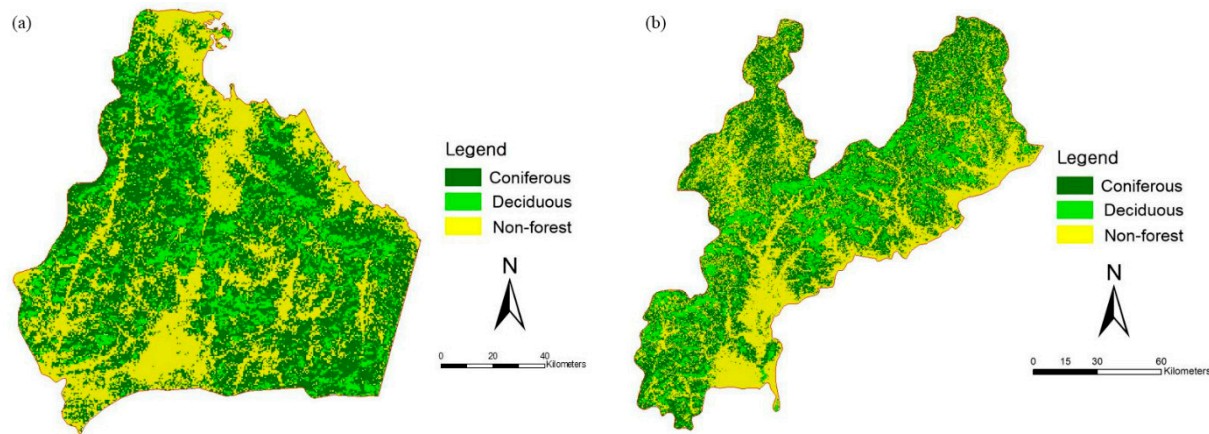

**Figure 9.** Forest types in the study area. (**a**) GWP; (**b**) SHP.

The coniferous forests in GWP had 44,454.9 ha (67.6%) burned, while the deciduous forests had 11,983.6 ha (18.2%) burned, with 32,471.3 ha more coniferous forest burned than deciduous forest (Figure 10a). In SHP, the area of burn in coniferous forests was 56,001.2 ha (55.7%), while that of deciduous forests was 18,822.9 ha (18.7%) (Figure 10b).

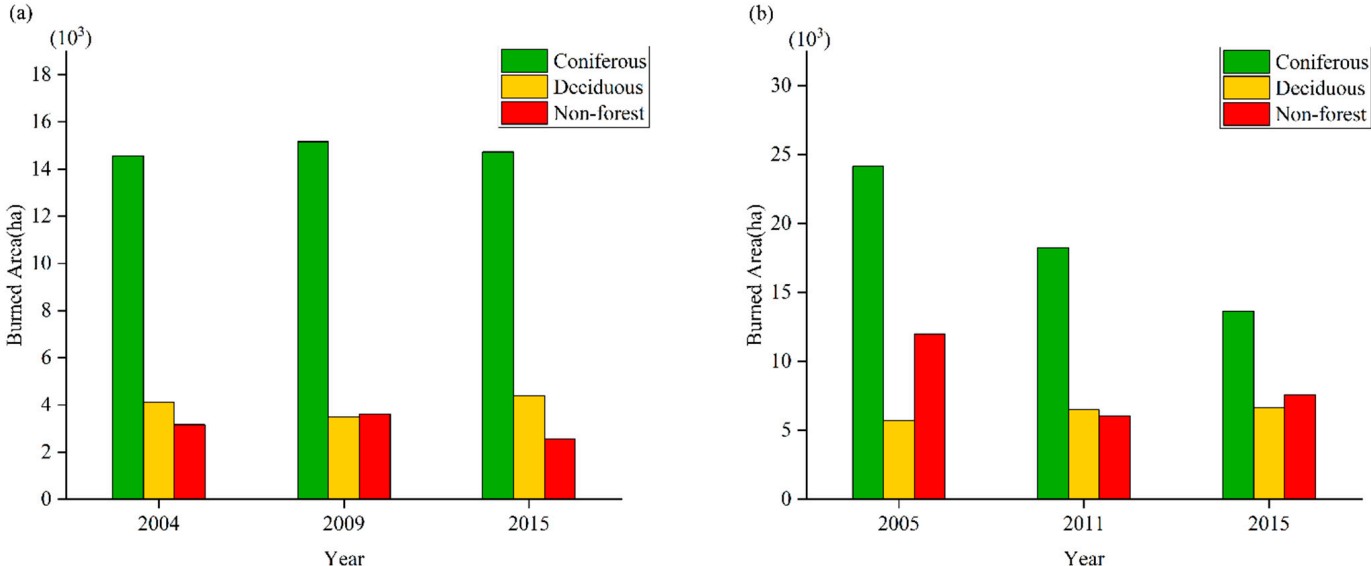

**Figure 10.** Areas of forest types burnt by fires in the study area. (**a**) GWP; (**b**) SHP.

To estimate the forest type with regard to burn severity, a forest map was derived from overlaid Landsat images and a burn severity map. It was investigated using five burn severity categories in four large fire-burned areas except for the "unburned burn severity level" because there was no damage in the forest. Overall results show differences in burn severity levels between coniferous and deciduous and coniferous and non-forest in large fires in GWP and SHP. While there was a difference in burn severity level between deciduous and non-forest in large fires in GWP, there was no difference between deciduous and non-forest in large fires in SHP. The non-forest area was greater in SHP than in GWP.

To investigate the difference in burn severity level between coniferous, deciduous, and non-forest areas in the four studied large fire-burned areas, the Wilcoxon paired test with a 5% significance level was used in this study. Table 4 shows differences in burn severity levels between coniferous and deciduous and between coniferous and non-forest in large fires in GWP and SHP. However, there was no difference in burn severity level between deciduous and non-forest in large fires in SHP. Non-forest area in SHP was more susceptible to large fire damage than in GWP.

**Table 4.** Wilcoxon test results for burned areas.

|  | Data Set | *p*-Value |
|---|---|---|
|  | Coniferous and deciduous | 0.02 |
| Gahak (2004) | Coniferous and non-forest | 0.02 |
|  | Deciduous and non-forest | 0.04 |
|  | Coniferous and deciduous | 0.03 |
| Hamju (2005) | Coniferous and non-forest | 0.03 |
|  | Deciduous and non-forest | 0.9 |
|  | Coniferous and deciduous | 0.02 |
| Jeongpyong (2011) | Coniferous and non-forest | 0.02 |
|  | Deciduous and non-forest | 0.18 |
|  | Coniferous and deciduous | 0.02 |
| Uiwon (2015) | Coniferous and non-forest | 0.02 |
|  | Deciduous and non-forest | 0.67 |

3.3.2. Topography

Elevation (Topography) in Burned Areas

In terms of elevation, Figure 11a shows the elevation of the burned area of the GWP. Overall, the elevation of fires tends to increase initially and then decrease. Most of the fires were in the 200–800 m range. It is worth noting that most burned areas in 2015 were in the 1100–1400 m range, which was different from other years, and the fires tended to occur at high elevations each year. The largest burned areas in 2004, 2009, and 2015 were in the 200–300 m range, the 500–600 m range, and the 600–700 m range, respectively. Figure 11b shows the elevation of burned areas in SHP. Most of the fires were in the 200–900 m range. The largest burned areas in 2005, 2011, and 2015 were in the 400–500 m range, the 300–400 m range, and the 600–700 m range, respectively.

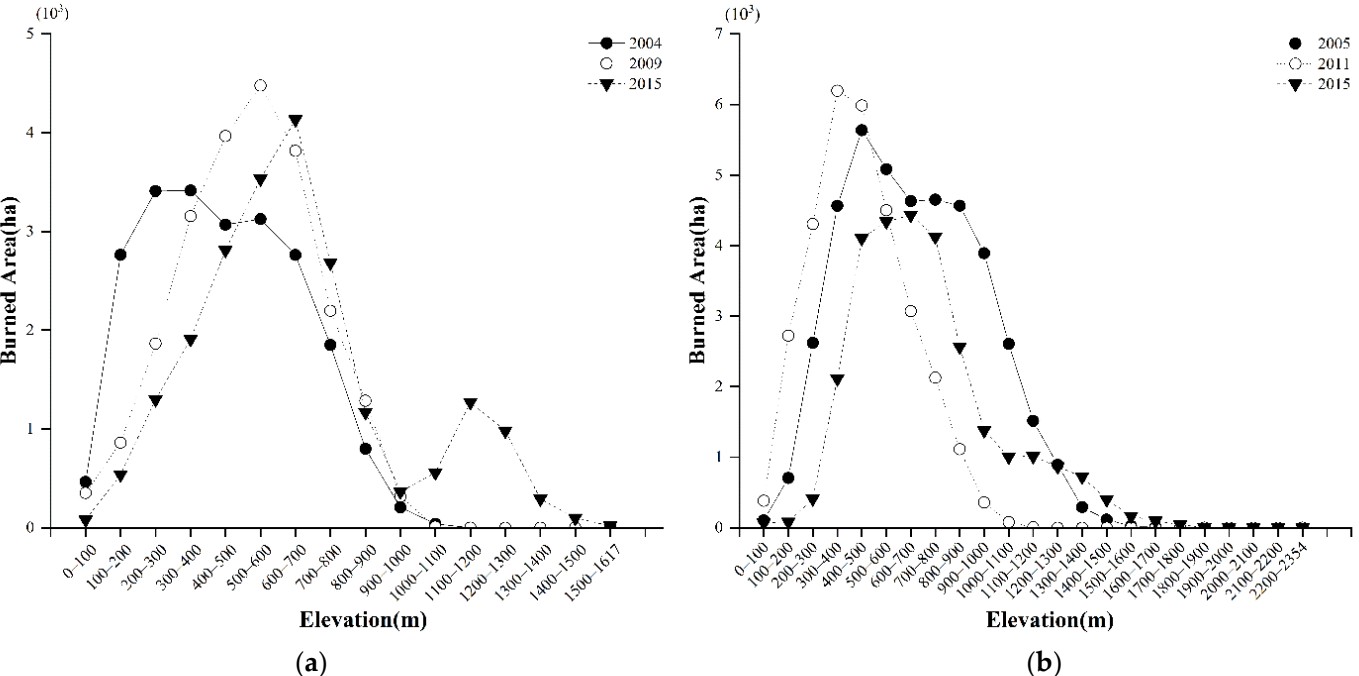

**Figure 11.** Elevations of burned areas in the study site. (**a**) GWP; (**b**) SHP.

Topography of Large Fire-Burned Areas

To investigate the topographic characteristics of four large fire-burned areas, elevation, slope, and landform were assessed.

As shown in Table 5, the largest elevation range was (=1459-83) 1376 m in the Hamju burned area, followed by Uiwon and Jeongpyong, and the smallest elevation range was 879 m in Gaehak. Furthermore, the highest average elevation in GWP was 615.9 m in the Uiwon burned area, followed by Jeongpyong and Hamju, and the lowest average elevation was 293.9 m in the Gaehak burned area. The highest average slope in SHP was 40% in the Hamju burned area, followed by Jeongpyong and Gaehak, and the lowest average slope was 37.2% in the Uiwon burned area.

Figure 12 shows the frequency of each landform type in the large fire-burned areas. There were no fires in shallow valleys, high ridges, or U-shaped valleys, and most large fires occurred on open slopes.

**Table 5.** Statistical results of elevation and slope in burned areas in both study regions.

| Name | Statistic | Elevation (m) | Slope (%) |
|---|---|---|---|
| Gaehak (2004) | Min. | 27 | 0 |
| | Max. | 906 | 138.5 |
| | Mean | 293.9 | 37.7 |
| | Range | 879 | 138.5 |
| | Std. D | 139.6 | 18.5 |
| Hamju (2005) | Min. | 83 | 0 |
| | Max. | 1459 | 181.4 |
| | Mean | 572.8 | 40 |
| | Range | 1376 | 181.4 |
| | Std. D | 245.4 | 17.5 |
| Jeongpyong (2011) | Min. | 38 | 0 |
| | Max. | 1107 | 159.1 |
| | Mean | 408.6 | 38.4 |
| | Range | 1069 | 159.1 |
| | Std. D | 187.3 | 17.8 |
| Uiwon (2015) | Min. | 187 | 0 |
| | Max. | 1399 | 130.5 |
| | Mean | 615.9 | 37.2 |
| | Range | 1212 | 130.5 |
| | Std. D | 168.8 | 16.5 |

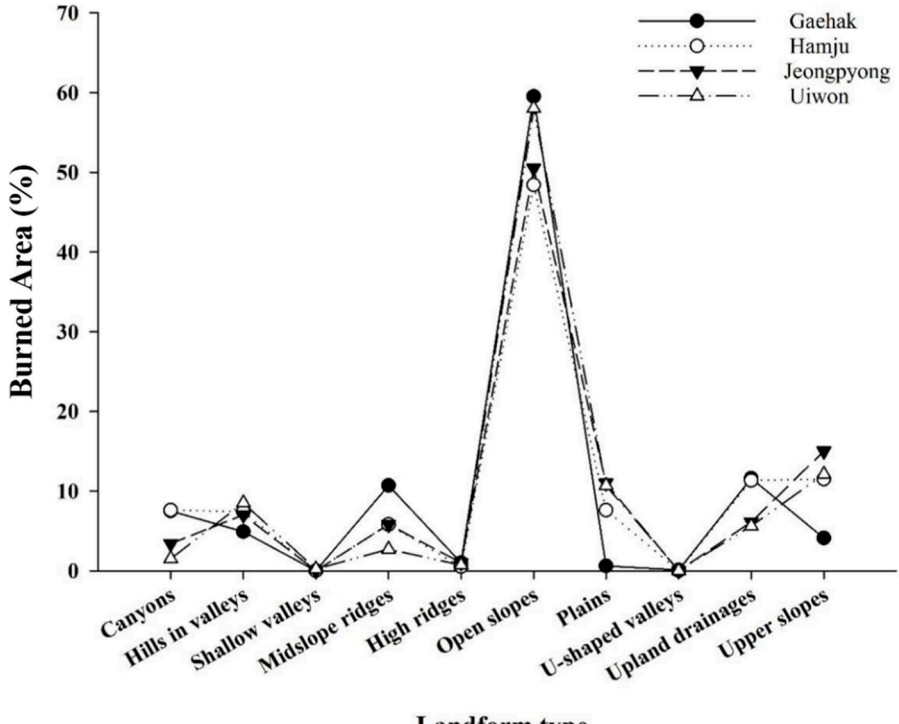

**Figure 12.** Distribution of landforms of the four burned areas in this study.

## 4. Discussion

### 4.1. Analysis of the Temporal and Spatial Distribution Characteristics of Fires

Temporally, the burned area of GWP showed a decreasing trend, and the burned area of SHP showed an increasing trend (Figure 5). Spatially, the burned area of GWP is widely distributed near non-forest areas, and the number of burned areas is larger, and the area is smaller, while the burned area of SHP is mostly distributed in forest areas and the number of burned areas is smaller, and the area is larger (Figure 6). From the perspective of fire sources, most of the fires in North Korea are caused by agricultural fires. It is easy to see

from the forest types of the two areas that the non-forest areas (which can be considered as cultivated land) of GWP are more scattered in distribution (Figure 9a), while the non-forest areas of SHP are more clustered (Figure 9b), and this difference leads to the different spatial distribution of fires in the two areas. From the perspective of fire suppression, North Korea lacks modern fire suppression conditions and still uses humans for fire suppression in most areas, so human accessibility is a key factor for fire suppression operations in North Korea. Most of the GWP is low in elevation, and non-forest areas (which can be considered as population concentrations) are widely distributed throughout the region (Figure 9a), so in case of a fire, it is easy to conduct fire suppression operations and quickly control the spread of the fire; the SHP non-forest areas are mostly clustered in patches on the southeastern side (Figure 9b), and during the spring fire-prone period, with the support of the southeastern monsoon, fires can easily spread to large forest areas on the northwestern side. Such fires are often difficult to extinguish due to low human accessibility and tend to form large burned areas.

### 4.2. Analysis of the Burn Severity of Large Fires

The burn severity of the four selected large fires showed that the large fire at Uiwon had the greatest "Extreme" burn severity level, showing the most severe burn severity of the four large fires in this study (Figure 8). From the locations of the four fires, only the large fire in Uiwon was surrounded by large areas of cultivated land and lacked water (Figure 7). From the remote sensing images before the fires, it can be found that there were small fire areas in the Uiwon area one year before the large fires, and these fire areas were very close to the cultivated land (Figure 7g), so it can be assumed that these are traces of agricultural fires. It can be inferred that such agricultural fires are common in the Uiwon area, but in previous years the fires did not spread over a large area, which to some extent led to the accumulation of combustible materials. When the fire spread in 2015, the accumulation of combustible materials from previous years increased the intensity of the fire and caused a more serious disaster.

### 4.3. Relationship between Forest Type, Topography, and Burn Severity

The results show that the burned area in coniferous forests is much higher than in deciduous forests (Figure 9). This is probably due to the fact that coniferous forests are prone to canopy fires, which are not easy to extinguish; under the effect of high wind, a fire usually spreads along the canopy and is extremely destructive. Thus, coniferous forests and their high levels of tree resin burned more easily. Furthermore, crown fires resulted in the complete destruction of many coniferous trees, in agreement with South Korean study results by Lee et al. [40] and Won et al. [41]. Deciduous forests usually experience surface fires, which generally burn ground cover plants, young trees, shrubs, shorter trees, bases of large trunks, and exposed roots. Although such fires spread quickly, the combustion is not the same as that in coniferous forests, is easier to extinguish, and produces relatively less damage.

The distribution of burned areas at different elevations shows that the elevation of fires tends to increase initially and then decrease, but the composition of the forest type will not change much over a short time. The highest elevation, mean elevation, and agricultural area elevation was higher in SHP than in GWP (Figures 9 and 11). Human activities, such as agricultural activities in high-elevation regions, can cause fires at higher elevations in mountainous areas of North Korea.

In terms of the topographic characteristics of the large fires, steep slopes are apparent in all four regions and are maintained because rainfall travels downslope. The persistent dry conditions foster combustion. The slope directly affects the moisture content of combustibles and has a large influence on heat propagation. Thus, convective heat and radiant heat intensity received by the combustibles increase, allowing the fire to spread at a faster speed. This is attributed to the "chimney effect" of concentrated heat energy in both provinces. The "chimney effect" is a natural phenomenon that occurs when the density

difference between a hot and a cold air column creates a natural flow, as through a chimney, or flow from lowland to highland regions. Therefore, large fires frequently occurred in steep slope areas in all four regions. Moreover, the topography was more rugged in the large fire-burned areas in SHP than in GWP, although there was no significant difference in slope between the four large fire-burned areas.

Fires easily spread on the windward-side open slopes in North Korean mountain topography, and open slopes are the most susceptible landform in North Korean burned areas (Figure 12). According to image interpretation and classification, windward-side open slopes are seriously impacted by fire. Of the three landforms where no fire was recorded, high ridges occupy a small portion of the total study area compared with other landform types, so the probability of occurrence was low; this is also true in shallow valleys. The U-shaped valley is a glacially eroded landform very rare in Korea, as shown by a classification of nearly 0%. Consequently, the Weiss [42] model needs to be modified for Korean topography.

## 5. Conclusions

In this study, fire trends in South Hamgyong Province (SHP) and Gangwon Province (GWP) in North Korea were analyzed using MODIS data. The burned areas were extracted based on the normalized burn ratio (NBR) index, and the spatial and temporal distribution characteristics of the fires were analyzed. Four large fires were selected to classify the burn severity based on the differential normalized burn ratio (dNBR) index. Forest types and topographic characteristics of burned areas were analyzed using forest types differentiated by the ISODATA clustering algorithm and digital elevation model (DEM) data. Finally, the relationship between forest type, topography, and burn severity was analyzed. After the results were compared and discussed, the following conclusions were derived.

The large-scale fires in the SHP are the main reason for the increasing trend in the overall burned area in the study area. Fires in the SHP are relatively concentrated along the BaekDu-DaeGan (BDDG), whereas fires in the GWP are scattered throughout the province. The proportion of burned area larger than 1000 ha is significantly larger in the SHP than in the GWP. Fires occur more frequently and are more destructive in the SHP than in the GWP. In terms of forest type, coniferous forest areas are more vulnerable to fire damage than deciduous forest areas because coniferous forests and their high content of tree resin are more likely to burn. With regard to large fire-burned areas in various landforms, most large fires occurred on wind-ward side open slopes, while there were almost no fires in shallow valleys, high ridges, or U-shaped valleys. It is believed that cultivation in high-elevation terrain and lack of fire extinguishing equipment and systems cause large fires to spread rapidly. South Korea, with a similar climate, is much less affected by wildfires. This is due to the initial fire alarm system and fire suppression equipment. In this respect, North Korea is very susceptible to large fire damage and must establish preparation measures.

This study was limited in its verification due to a lack of ground data. In addition, wind data were not used for fire analysis due to the absence of data from North Korea. For the next investigation, more study areas, such as North Hamgyong Province and Nahngrim Mountain, need to be investigated.

**Author Contributions:** Conceptualization, R.J. and K.-S.L.; methodology, R.J. and K.-S.L.; software, R.J.; validation, R.J. and K.-S.L.; formal analysis, R.J.; investigation, R.J.; resources, K.-S.L.; data curation, R.J.; writing—original draft preparation, R.J.; writing—review and editing, K.-S.L.; visualization, R.J.; supervision, K.-S.L.; project administration, K.-S.L.; funding acquisition, R.J. and K.-S.L. All authors have read and agreed to the published version of the manuscript.

**Funding:** Basic Research Project of the National Research Foundation of Korea (2013R1A1A2010007), Samsung Academic Research: S-2012-0796-000-1, National Nature Science Foundation of China (41807508), Jilin Provincial Science and Technology Department Project (20200403030SF).

**Conflicts of Interest:** The authors declare no conflict of interest.

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
