# Peer review of "Investigation of Forest Fire Characteristics in North Korea Using Remote Sensing Data and GIS"

_remotesensing, doi:10.3390/rs14225836_

Round 1

Reviewer 1 Report

General comments:

The goal of this study was to examine the spatial and temporal trends of forest fires in two North Korean provinces, a severely understudied region with an abundance of fires that present significant social and economic risk to inhabitants.

Major comments:

The Conclusions section of this paper is underdeveloped and offers little to no context for the findings of the study, nor does it adequately address the limitations of their design or utility of their findings. There are some areas of the Results that could be moved to the Conclusions, or perhaps the authors could restructure the manuscript to have a “Results and Conclusions” in the same section, but regardless there is significant work needed in this area.

Minor comments:

The over-use of abbreviations (e.g. FFDA for forest fire damaged area, FF for forest fire) makes the manuscript almost impossible to follow in many places and also is not stylistically in line with most publishing in this field. Also it is grammatically incorrect to start a sentence with an abbreviation, so if you’re writing the sentence “NK is very susceptible…” it needs to be “North Korea is very susceptible…” even if you use the abbreviation “NK” elsewhere.

I suggest the following changes for these abbreviations:

FF – use forest fire the first time then simply use “fire” or “wildfire”

LFF – “large fire” or “large wildfire” (you also need to define what you mean by “large” and why you’re referring to some as forest fires and others as large forest fires, unless you’re making a distinction between the two t0hat is important for the analysis just call them all forest fires and define in the Methods if you had a minimum size)

FFDA – “burn scar” or “burned area”, in many cases you could simply replace this with “fire” too since you’re often using them to mean the same thing

RS – “remote sensing” don’t abbreviate this

BS – “burn severity” or “severity”

LFFDA – There is no reason why you need to differentiate this from other burned areas, simply use “burn scar” or “burned area” as suggested above

Line comments:

Line 40: Should be plural “Forest fires…” and again do not abbreviate forest fires, just call them “fires” from here on out and the reader will know you mean forest fires. Also, this opening line is so broad and vague that it ends up not being true, there are many areas where forest fires are beneficial in the long run for the plants and animals in that area. I suggest rewriting the line.

Line 52: “it occurred about 170 times of FF just in mid-may occurred in NK” This is not grammatically correct and I’m a little confused what the sentence is trying to say? I think maybe “… by mid-May alone 170 fires occurred in North Korea”? However that also needs to be specified if that’s an average over multiple years or if that’s for one particular year?

Methods, study site description: It would be very helpful to have some seasonal or even annual averages for temperature and precipitation here, simply saying it is “warm and rainy” can mean vastly different things to international readers.

Methods: I am unclear how your hotspots count (in Figure 4) is differentiating between controlled agricultural burns and forest fires that are not controlled? Because in your introduction you set this study up as investigating damaging forest fires, so you need to somehow be eliminating intentional agricultural burning. Unless you are making the argument that agricultural burning is damaging? Either way you need to more explicitly clarify how you dealt with this or why it is justified to not separate them.

Figure 6: You have multiple consistency issues for this figure. In panels b and d you use “Damaged Area” as the y-axis whereas in Figure 5 you used “Burned Area”. I suggest using “Burned Area” for both as it is more accurate and in line with the common usage in the literature. Also, it would be very helpful to the reader if you made the bar fill patterns consistent across panels; it’s very confusing to have the same category be a different fill type. Finally, you define the categories in hectares in line 266 but on your figure you use a mix of hectares and km2? Please be consistent and use hectares for the figure so that your categories are the same!

Line 275: “the amount of FF” should be “the number of FF”

Line 277: I’m unclear what “concentration ratio and magnitude” mean here, do you mean to say that there were more fires per km2 and those fires tended to be larger in size? Because if that’s the case I suggest you simply say that.

 Line 286: “it is considered that the flames leaped to other parts of the LFFDA in Uiwon” I do not know what this is trying to say? The patchy nature of burn severity is expected and well-documented for fires occurring in forests, and that patchy nature is due to factors such as fuel moisture, weather at the time of burning, differences in dominant vegetation, differences in vegetation structure, slope, aspect, etc. It is NOT due to “flames leaping to other parts” of the landscape.

Figure 7: I see a fair amount of cloud cover that appears to overlap your burned area for the Uiwon-gun Fire, however it does not appear that you masked out that cloud cover before calculating NBR and dNBR? You have to mask out large bodies of water and all cloud cover before you calculate those or you will get anomalous results. Also, the panels for burn severity (bottom row of the figure) are so small that the legends are rendered completely unreadable.

Line 338: Why are you suddenly citing other studies here in the Results? That would fit better in the Discussion I think.

Figure 11:  I’m not sure this figure is the best way to present these results, I think you could simply report them in the text and it would be a lot more clear.

Line 362: This whole paragraph gets a little buried in numbers and difficult to read, I suggest you simply talk about general trends and any anomalous years, then let the reader look at the figure if they’re interested in each individual year.  

Figure 13: I suggest you increase the font size of the text in your legend, it is very small and rather difficult to read. However, I’m not sure this figure is really necessary in general since you’re effectively showing that these four fires were virtually the same? You could simply say that in the text and remove this figure.

Conclusions: It’s generally good to re-define your abbreviations at their first use in the Conclusions section, so rather than just using GWP actually say “Gangwon Province (GWP)”

Author Response

Dear reviewers:

We are very grateful to your comments for the manuscript. According with your advice, we tried our best to amend the relevant part and made some changes in the manuscript. These changes will not influence the content and framework of the paper. All of your questions were answered below. In the attachment, we list the modifications. And the revision marks were retained in the revised paper.

We appreciate for Reviewers’ warm work earnestly, and hope that the correction will meet with approval. Should you have any questions, please contact us without hesitate. 

Once again, thank you very much for your comments and suggestions.

Yours Sincerely,

Ri Jin

Reviewer 2 Report

The article presents a very interesting interdisciplinary topic with elements of remote sensing, environmental protection, fire engineering, and geoinformatics. The authors undertook the analysis of fires in a politically challenging area, which significantly limited access to data. Nevertheless, the conducted research showed very interesting results. The proposed methodology turned out to be effective in identifying forest fires. The selection of references is very accurate and essential, supported by a concise introduction to the article. The description of the methodology is more detailed and reproducible. I think the authors should work more with the figures. Some of them require a few necessary corrections.
A few questions:
- How do you differentiate forest fire or other fires? For example, fires of meadows, wastelands, and field fires.
- Do you know anything about the existence of some hidden (non-public) database of fires in North Korea?
I put a few minor remarks in the PDF file.

Author Response

(The authors gave the same response as above.)

Round 2

Reviewer 1 Report

Major comments: The authors have obviously done a lot of additional work on the manuscript and I appreciate the changes they have made! However, while the Discussion section is now longer I’m not sure it’s actually that much more fleshed out as a Discussion section because it reads mostly as if it is simply an extension of the Results section. I think there is still a lot of opportunity for the authors to place the results within the context of other existing research in similar forest types, rather than simply summarizing their results. However, section 4.3 is well written and does actually incorporate existing research, so this section is I think a good model for what the rest of the Discussion could be.

Line 12: Change “forest fire” to “forest fires”

Table 3: I think this table may be unnecessary, since all of the information contained in the table is found in the paragraph text. I suggest that you remove the table.

Author Response

(The authors gave the same response as above.)
